# Transcriptomic analysis of immune cells in a multi-ethnic cohort of systemic lupus erythematosus patients identifies ethnicity- and disease-specific expression signatures

Gaia Andreoletti[1], Cristina M. Lanata [2], Laura Trupin[2], Ishan Paranjpe[1], Tia S. Jain[1], Joanne Nititham[2], Kimberly E. Taylor[2], Alexis J. Combes [3,4,5], Lenka Maliskova[2], Chun Jimmie Ye [4,6], Patricia Katz[2], Maria Dall'Era[2], Jinoos Yazdany[7], Lindsey A. Criswell[2] & Marina Sirota [1✉]

Systemic lupus erythematosus (SLE) is an autoimmune disease in which outcomes vary among different racial groups. We leverage cell-sorted RNA-seq data (CD14+ monocytes, B cells, CD4+ T cells, and NK cells) from 120 SLE patients (63 Asian and 57 White individuals) and apply a four-tier approach including unsupervised clustering, differential expression analyses, gene co-expression analyses, and machine learning to identify SLE subgroups within this multiethnic cohort. K-means clustering on each cell-type resulted in three clusters for CD4 and CD14, and two for B and NK cells. To understand the identified clusters, correlation analysis revealed significant positive associations between the clusters and clinical parameters including disease activity as well as ethnicity. We then explored differentially expressed genes between Asian and White groups for each cell-type. The shared differentially expressed genes across cells were involved in SLE or other autoimmune-related pathways. Co-expression analysis identified similarly regulated genes across samples and grouped these genes into modules. Finally, random forest classification of disease activity in the White and Asian cohorts showed the best classification in CD4+ T cells in White individuals. The results from these analyses will help stratify patients based on their gene expression signatures to enable SLE precision medicine.

[1] Bakar Computational Health Sciences Institute, University of California, San Francisco, CA, USA. [2] Russell/Engleman Rheumatology Research Center, Department of Medicine, University of California San Francisco, San Francisco, CA, USA. [3] Department of Pathology, University of California San Francisco, San Francisco, USA. [4] ImmunoX Initiative, University of California San Francisco, San Francisco, CA, USA. [5] UCSF CoLabs, University of California San Francisco, San Francisco, CA, USA. [6] Institute for Human Genetics, University of California San Francisco, San Francisco, CA, USA. [7] Division of Rheumatology, Department of Medicine, University of California San Francisco, San Francisco, CA, USA. ✉email: marina.sirota@ucsf.edu

Systemic lupus erythematosus (SLE) is a complex multisystem autoimmune disorder characterized by dysregulation of the innate and adaptive arms of the immune system[1]. The predisposition and clinical phenotype of SLE, a disease associated with compelling morbidity and mortality, are attributed to a combination of genetics, hormones, and environmental factors[2]. Heterogenous clinical and serologic manifestations, a waxing and waning course, and delays in diagnosis contribute to the complexity of this disease.

Genome-wide association studies have discovered ~100 susceptibility loci for SLE[3]. Many of these loci are unique to European-Americans patients while others have been identified in African American or Hispanic-Americans[4,5]. Susceptibility to SLE has a strong genetic component, and trans-ancestral genetic studies have revealed a substantial commonality of shared genetic risk variants across different genetic ancestries that predispose to the development of SLE[6]. More than half of patients with SLE show dysregulation in the expression of genes in the interferon (IFN) pathway[7]. Scientific evidences have identified that the dominant group most commonly diagnosed with SLE are women of minority status in low socio-economic environments[8]. A recent epidemiologic study comparing lupus manifestations among four major racial and ethnic groups found substantial differences in the prevalence of several clinical SLE manifestations among racial/ethnic groups and discovered that African Americans, Asian/Pacific Islanders, and Hispanic patients are at increased risk of developing several severe manifestations following a diagnosis of SLE[9]. It is believed that an increased genetic risk burden in these populations, associated with increased autoantibody reactivity in non-white individuals with SLE, may explain the more severe lupus phenotype[6]. As patients with similar Systemic Lupus Erythematosus Disease Activity Index, or SLEDAI[10], scores may have different prognoses and treatment responses[11], there is an urgent need to establish a new method of stratification of lupus patients.

With the recent advances in molecular measurements and computational technologies, there are incredible opportunities to characterize disease-associated genes and pathways. Previous studies have used machine learning (ML) and clustering approaches to try to stratify patients with SLE based on different parameters including clinical data, expression quantitative trait loci (eQTLs), methylation, and transcriptomic data[12–15]. However, these studies have been mainly conducted on whole blood or peripheral blood mononuclear cells (PBMCs), therefore not considering the involvement of different immune cell types in disease, and in White cohorts, thus without taking into account the relationship between disease activity and ethnic background.

In our own previous work, unsupervised clustering of the 11 American College of Rheumatology (ACR) classification criteria in an ethnically diverse lupus cohort revealed three stable clusters, characterized by significant differences in several SLE features, as well as the lupus severity index[12]. Following up on the clinical clustering study, the objective of this study which leverages the same cohort is to use large-scale transcriptomic data from diverse ethnic populations to identify transcriptomic signatures in SLE relating to various clinical and demographic factors and better sub-classify SLE patients into more clinically actionable groups. More specifically, we leverage bulk RNA sequencing data on four immune-cell types sorted from peripheral blood mononuclear cells (PBMCs) (CD14+ monocytes, B cells, CD4+ T cells, and NK cells) of 120 patients (63 Asian and 57 White individuals) from a multi-racial/ethnic cohort of individuals with physician-confirmed SLE. This study aims to look for specific-transcriptomic effects in each of these immune cell types using a four-tier approach: unsupervised clustering, differential expression analyses, gene co-expression analyses, and machine learning approaches (Fig. 1).

Our approach helped to identify patient sub-clusters based on their gene expression data and to describe the involvement of CD14+ monocytes, B cells, CD4+ T cells, and NK cells in disease in an SLE multiethnic/racial cohort. This classification might provide insightful information to improve predictions of disease outcomes and subtype-specific mechanistic pathways that could be strategically targeted. We further explore molecular pathways that underlie the clinical and demographic differences in SLE.

## Results

**Unsupervised K-means clustering identified specific patient clusters per immune cell-type.** After profiling 120 SLE patients for cell sorted bulk RNA-seq data (CD4+ T cells, CD14+ monocytes, B cells, and NK cells) from a multi-racial/ethnic cohort, a total of 10% of the samples were removed after QC filtering retaining 415 samples (91 NK, 105 B cells, 108 CD4+ T-cells, and 111 CD14+ monocytes) (Supplementary Fig. 1A). Batch effects were taken into account in all the analyses using the limma software. Using the QC'ed data, we wanted to look for specific-transcriptomic effects in each of these cell types using a four-tier approach which included: unsupervised clustering; differential expression analyses, gene co-expression analyses, and machine learning (Fig. 1).

Unsupervised K-means clustering on the individual cell types was utilized to identify patient sub-clusters. Clusters with a Jaccard mean stability score greater than 0.6 were considered stable and therefore only clusters with a score greater than or equal to 0.6 were retained for further analyses (Supplementary Data 3). Clustering on CD4+ T cells and CD14+ monocytes yielded three distinct clusters, while only 2 clusters were identified for B cells and NK cells (Fig. 2A). We then sought to test for association between the transcriptomic K-means clustering and the clinical and demographic parameters (Fig. 2B).

We observed in CD4+ T cells a positive association with lupus severity index[16], SLEDAI activity score[10,17], ACR criteria thrombocytopenia, and ACR criteria Oral Ulcers. CD4 transcriptomic cluster 1 was comprised of 44 patients and was characterized by a high prevalence of photosensitivity, seizures, younger age at diagnoses, and presence of anti-Smith autoantibody positivity (Supplementary Data 4). Interestingly, we detected a positive association with CD4+ T cells cluster 1 and lupus severity index, which is a validated tool based on weighted scores of the ACR classification criteria[16]. The ACR criteria positively associated with CD4+ T cells cluster 1 were history of seizure and thrombocytopenia. There was also a positive association with this cluster and the SLICC score, which tracks organ damage over time, likely associated with disease severity. The same cluster was negatively associated with age at diagnoses, ACR leukopenia, ACR psychosis (Fig. 2B).

CD4+ T transcriptomic cluster 2 was positively associated with age at diagnoses, SLEDAI score, and ACR leukopenia. The same cluster was negatively associated with ACR seizure. Patients with a high prevalence of malar rash, psychosis, anti-dsDNA autoantibody positivity, and SLEDAI score constitute the CD4 transcriptomic cluster 2 (Supplementary Data 4). Finally, CD4+ T transcriptomic cluster 3 was negatively associated with ACR malar rash and the presence of anti-Smith autoantibody positivity.

In CD14+ monocytes we observed a significant positive association in cluster 1 with race White, as well as a negative association with race Asian (Fig. 2B). Moreover, cluster 2 of the CD14+ monocytes was negatively associated with race White. This was the only association observed for cluster 2. On the clinical level, cluster 1 of CD14+ monocytes consisted of 29 patients and it was defined by lower rates of flare, and anti-dsDNA autoantibody positivity, while cluster 2 encloses 58 patients characterized by high

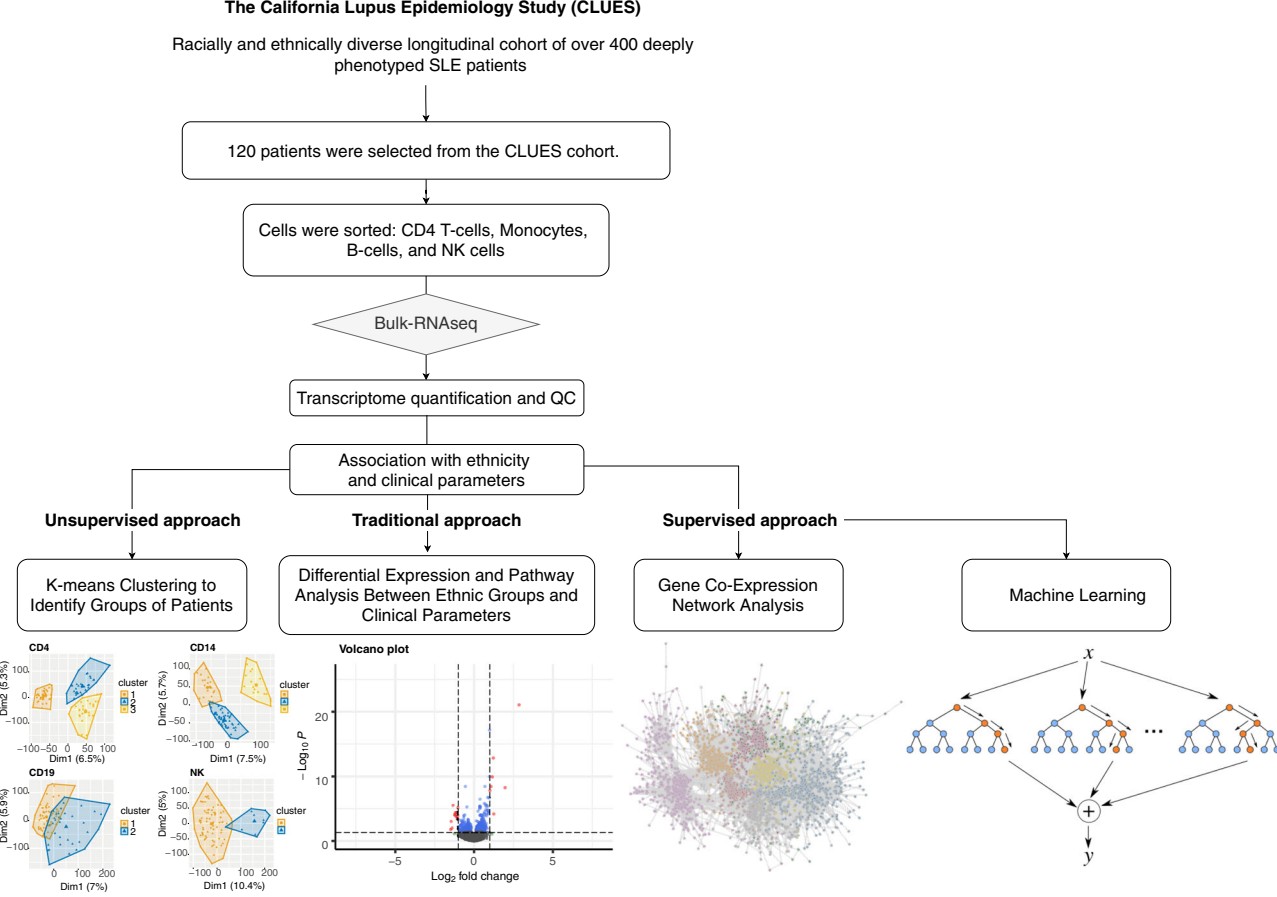

**Fig. 1 Targeted tier approach.** 120 patients were selected for bulk RNA-seq analyses from the CLUES cohort (Supplementary Data 1). Data were QC'ed and processed for transcriptome quantification (Supplementary Fig. 1). After removal of the outliers/low-quality samples, we used three types of approaches: an unsupervised approach, a more traditional approach, and supervised approaches.

rates of flare severity (measured as mild 1, moderate 2, severe 3) (Supplementary Data 5).

The two clusters of CD19, B cells, showed for cluster 1 a negative association for SLICC score, lupus severity index, and ACR seizure, while on the other hand for cluster 2 we observed a positive association for SLICC score, lupus severity index, and ACR seizure. (Fig. 2B). Cluster 1 of B cells comprised 68 patients with a low prevalence of seizure, lymphopenia, and the low score for lupus severity index and SLICC score, which suggested low disease activity. On the other hand, cluster 2 of B cells ($n = 38$) featured patients with a high prevalence of seizure, lymphopenia, and a high score for lupus severity index and SLICC score, suggesting high disease activity (Supplementary Data 6).

In the two clusters of the NK cells, we observed a significant association between the presence of antinuclear antibody and cluster 1, as well as a negative association between antinuclear antibody and cluster 2. No further associations were seen between the clinical and demographic parameters in the two clusters in NK cells (Fig. 2B, Supplementary Data 7).

Interestingly, from this correlation analysis, we noted a similar relationship between the transcriptomic clusters of CD4+ T cells and the clinical K-means clusters previously identified by our group solely using ACR clinical criteria[12] (Fig. 2C). Supporting the similarity between clusters identified in the CD4+ T cells and the clustering using uniquely clinical features we identified an important overlap in patient membership (Fig. 2C). A total of 61% of the individuals in the CD4+ T cell transcriptomic cluster 1 were also present in the severe cluster of the clinical data. Finally, as also

observed in the clinical clusters, we statistically confirmed the association with CD4+ T cells transcriptomic clusters and lupus severity index ($p < 0.05$) (Fig. 2D). Overlaying this information allowed us to look at a global relationship between clinical data and the transcriptional profiles across the cell types. In addition, as we detected an association between CD14+ monocytes and ethnicity, we used DESeq2 to identify the differentially expressed genes between SLE patients from different ethnic backgrounds.

**Differences in gene expression levels within each cell type significantly differ between ethnic populations**. Differential expression analysis on demographic and clinical features identified a few significantly (FDR < 0.05 and abs log2FC > 1) differentially expressed genes. Specifically, 29 genes were identified for the DE on lupus severity index, 3 on SLICC score, 5 for DE on SLEDAI score, 9 for DE on ACR criteria for lupus nephritis (Supplementary Data 8). Pathway analyses derived from differential expression analysis on the three different clinical indexes and the presence of lupus nephritis, a type of kidney complication caused by SLE, across the immune cell types showed enrichment of pathways in CD4 and CD14 for lupus severity index and SLICC score while SLEDAI score was enriched in CD4 only (Supplementary Fig. 6).

Remarkably, a larger number of differentially expressed genes between Asian and White groups were identified for each cell-type (Fig. 3A, Supplementary Figs. 2–5). In the CD4+ T cells for the White cohort, the Actin Filament Associated Protein 1 (*AFAP1), USP32P1*, and, *NAP1L3* (Nucleosome Assembly Protein

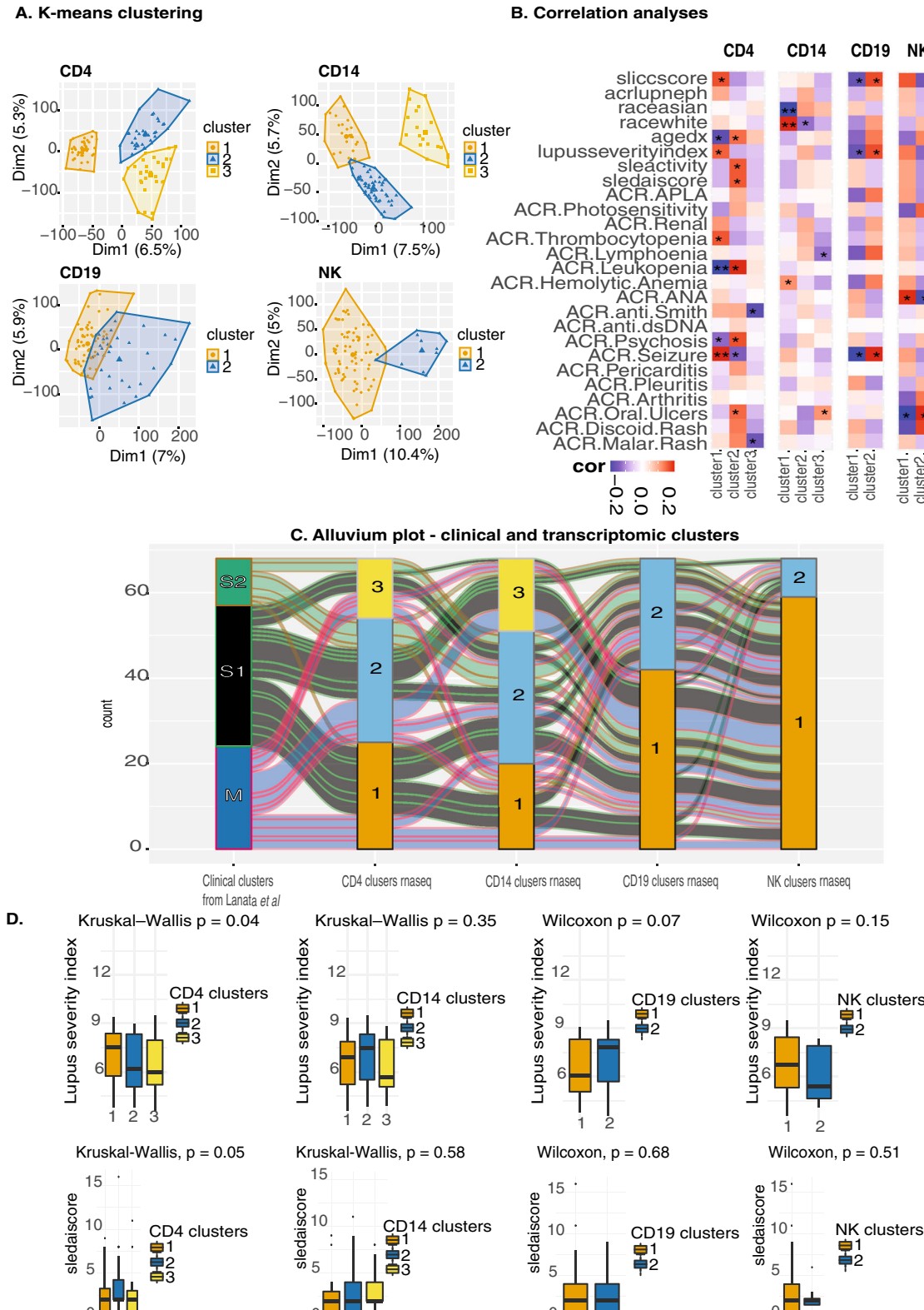

**Fig. 2 K-means clustering analyses and correlation between transcriptomic clusters and clinical parameters including disease activity and ethnicity.**
**A** K-means clustering on the CD14+ monocytes, CD4+ T-cells, B cells, and NK cells. **B** Association analyses between clinical parameters and K-means s
clustering. Red–positive association, blue–negative association. The number of stars indicates the level of significance. **C** Alluvium plot visualizing the
distribution of the samples according to different clusters. **D** Distribution of lupus severity index and SLEDAI score across clusters with *p*-value computed
using non-parametric tests.

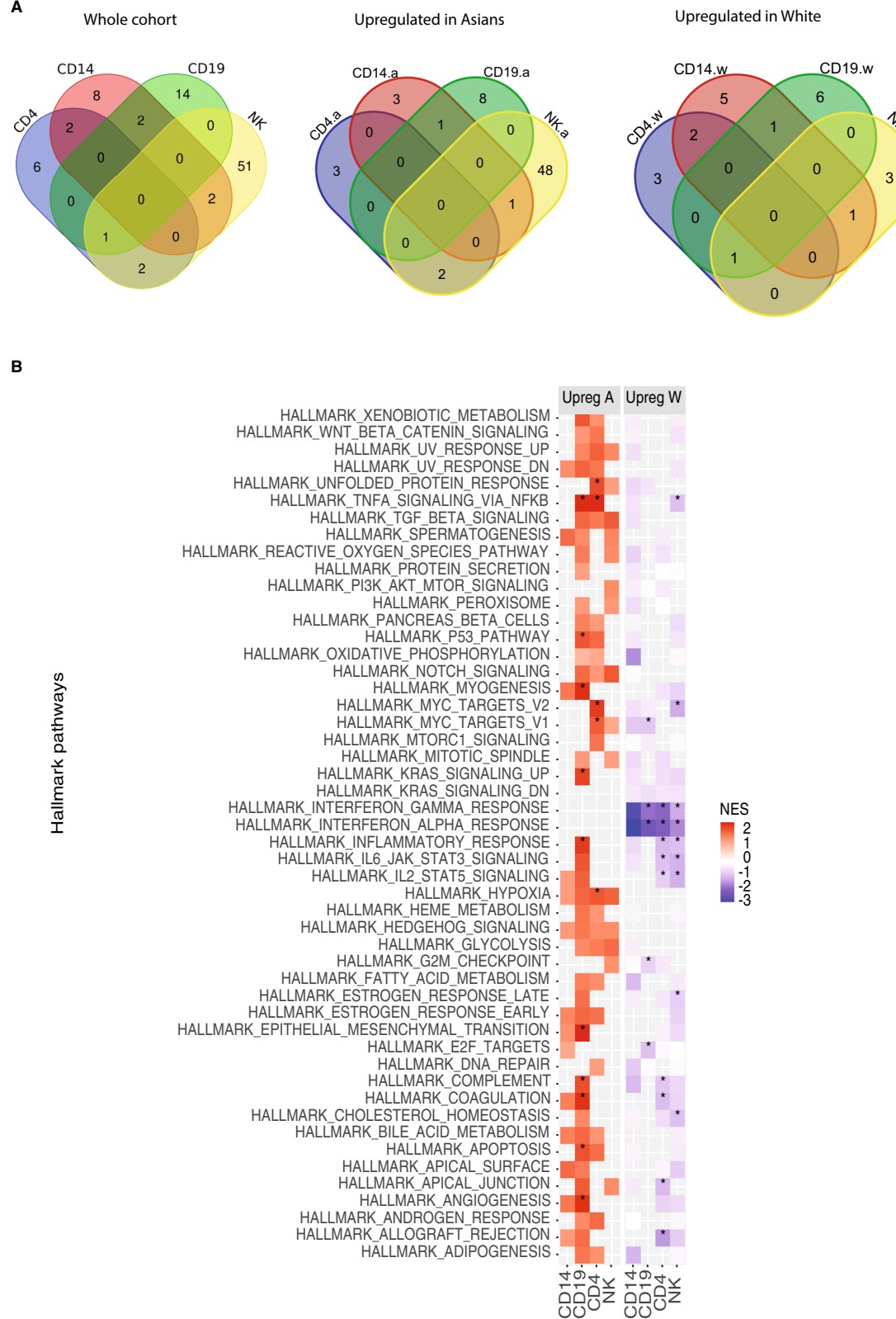

**Fig. 3 Differential gene expression analyses. A** Identification of the common DE genes (padj < 0.05 and abs log2FC >1) White vs. Asian cohorts across the different cell types. Supplementary Data 2a, b, and 2c list the genes shared and specific for each cell type. **B** Pathway enrichment analyses for the associated genes in White vs. Asian cohorts.

1 Like 3) were the top three significantly upregulated (padj < 0.05) genes. ARHGEF10 and FMN1 were the top significant genes in CD4$^+$ T cells upregulated in Asians. An example of significantly upregulated genes in CD14$^+$ monocytes cells for the White cohort was SNORD3B-2 and the lncRNA maternally expression gene 3, MEG3. In Asians, Neurotensin Receptor 1 (NTSR1), and RETN were among the top significantly upregulated genes in CD14$^+$ monocytes. Top upregulated genes in B cells for the White cohort were ARHGAP24 and CHL1 (Cell Adhesion Molecule L1 Like). Genes encoding for immunoglobulin heavy-chain and light-chain genes were statistically significantly upregulated in Asians in B cells. Regarding NK cells, examples of upregulated significant genes in Whites were SNORD3B-2, and Cytokine Like 1 (CYTL1), while an example of upregulated genes in Asians was HPGD and ADAM19. Pathway analyses in Asians revealed upregulated pathways involved in metabolism and transcriptional activity in CD14$^+$ monocytes and CD4$^+$ T cells whereas in Whites, both B cell and NK cells were enriched for upregulated pathways for integrin and IL2 signaling indicating a higher activity of specific cell types in Asians and Whites, respectively (Fig. 3B). There were no genes shared across all four cell types, (Supplementary Data 2) and the only RPL3P2 is shared across CD19, CD4, and NK cells.

**Gene co-expression analyses within ethnic populations and disease activity revealed specific modules of genes.** Classical approaches to analyze the transcriptome data by using differential gene expression analysis based on sample groups defined by a selection of clinical parameters precluded dissection of the heterogeneity of SLE. Co-expression analysis, on the other hand, identifies similarly regulated genes across samples, and then groups these genes into modules, which can then be explored for each patient sample individually or for entire patient groups[18,19]. As we found the largest number for DE genes on ethnicity, we decided to use CoCena2 on the patients' data grouped by ethnic background combined with disease activity (SLEDAI score). Hence, patients were divided into White-high, Asian-high (high disease activity defined by a SLEDAI score >= 6) from White-low, Asian-low (low disease activity defined by a SLEDAI score <6) for each cell type. The CoCena pipeline was run on each cell line independently. The pipeline identified 8, 5, 10, and 4 co-expression modules for CD4$^+$ T cells, CD14$^+$ monocytes, CD19, and NK cells, respectively. Analysis of the modules revealed group-specific enrichment of co-expressed gene modules (Fig. 4A). Enrichment analyses on each of the modules identified associated gene signatures displaying distinct functional characteristics, which distinguish the different sample groups. (Fig. 4B, Fig. 5). We then annotated the modules according to their behavior into seven categories: ethnicity-specific independent of disease activity, opposite behaviors according to ethnicity, White specific, Asian specific, low disease activity-specific, high disease activity-specific, and high disease activity specific for White. For example, the IL6 JAK STAT3 signaling pathway was identified as having opposite behavior in Asian and Caucasian populations and was enriched in the modules dark gray and gold for CD4$^+$ T cells. More precisely the dark gray module was highly expressed in Asian patients with low disease activity and in White patients with high disease activity. The maroon module of CD14$^+$ monocytes was annotated as being specific for low disease activity as it was low expressed in Asians and White individuals with high SLEDAI scores. Interestingly, seven of the 10 modules in B cells were highly expressed in Whites compared to Asians regardless of their disease activity, while three of the four modules in the NK cells were exclusively highly expressed in Asian patients with high disease activity (Fig. 4A). An extended analysis of the modules

revealed a prominent enrichment of inflammatory, autoimmune, and interferon pathways (Fig. 5, Fig. S7).

**Machine Learning approaches predicted SLE disease activity in individual ethnic groups.** Based on the results of the previous analyses and on the clustering of the patients according to co-expressed modules found on ethnicity and disease activity, we decided to use random forest (RF) on the transcriptomic data for each cell type within each racial group to discriminate between patients with high disease activity (SLEDAI score >=6) and low disease activity (SLEDAI score <6). Receiver operating characteristic (ROC) analysis of each RF model showed an AUC of 0.8 (Fig. 6A), indicating the consistent efficiency of the model in discriminating high disease activity samples from low disease activity samples in CD4$^+$ T cells and CD19 cells when the whole cohort is taken into account. With regards to the White cohort, the RF model trained on CD4$^+$ T cells and CD14$^+$ monocytes showed an AUC of 0.71 and 0.79, respectively. On the other hand, the models trained on the Asian cohort showed an AUC higher than random only in CD4 T cells (AUC 0.62). The models trained with the top feature of the other ethnic background showed an AUC less than 0.6 which indicates inefficiency in the model in discriminating patients based on disease activity and underlaying the ethnicity-specific features for each model (Supplementary Fig. 8).

The top contributing features identified by the random forest algorithm to segregate high and low disease activity patients in CD4$^+$ T cell in the White cohort were EPSTI1 and LAIR1, while in the Asian cohort these were: IRS1, RRM2, ATP11C, ACCS, and CCR6. With regards to the B cells, the top features for the White cohort were SDK2, IGKV2D, LAIR1, DSP, and IGHV3 while the top features for the Asians cohort were: CCFC292, SUDS3, DUSP4, and HSPB1 (Fig. 6B). Importantly some of these genes have been previously associated with SLE. We observed that the model was able to select SLE-relevant features. In summary, these results suggested the potential interaction between clinical features and potentially disease-related genes, which could help explain the multifactorial, heterogeneous, and systemic nature of the disease within White and Asians.

**Discussion**
SLE has a great diversity of presentation and treatment responses. There has been considerable progress in genetic studies of SLE, thanks partly to technological advances and to confirming previously reported genes. Due to the complexity of the pathogenesis and the heterogeneous nature of SLE clinical manifestations, imprecise diagnosis and poor treatment efficacy remain two major obstacles that drastically affect the outcome of patients with SLE. Here we profiled a multiethnic cohort of 120 patients with SLE using cell sorting RNA-seq data (CD4$^+$ T cells, CD14$^+$ monocytes, B cells, and NK cells). Since SLE severity is known to vary widely between racial and ethnic groups, analysis of a large multiethnic cohort is crucial for understanding the genetic and non-genetic determinants of this ethnic-associated variability.

With an unsupervised clustering approach, we were able to unbiasedly stratify patients based on gene expression signatures allowing the identification of separate molecular pathways underpinning disease in SLE for each cell type (Fig. 2A). Transcriptomic cluster 1 of the CD4$^+$ T cells has a lower age of onset (mean 24 y/o) compared to cluster 2 (31 y/o) and cluster 3 (29 y/o) of the same cell type (Supplementary Data 4) suggesting an enrichment of higher disease severity in cluster 1 compared to cluster 2 in CD4$^+$ T cells. This is also supported by the alluvium plot showing a higher clustering of severe patients, according to k-means clustering on only clinical criteria, in the cluster 1 and 2 of

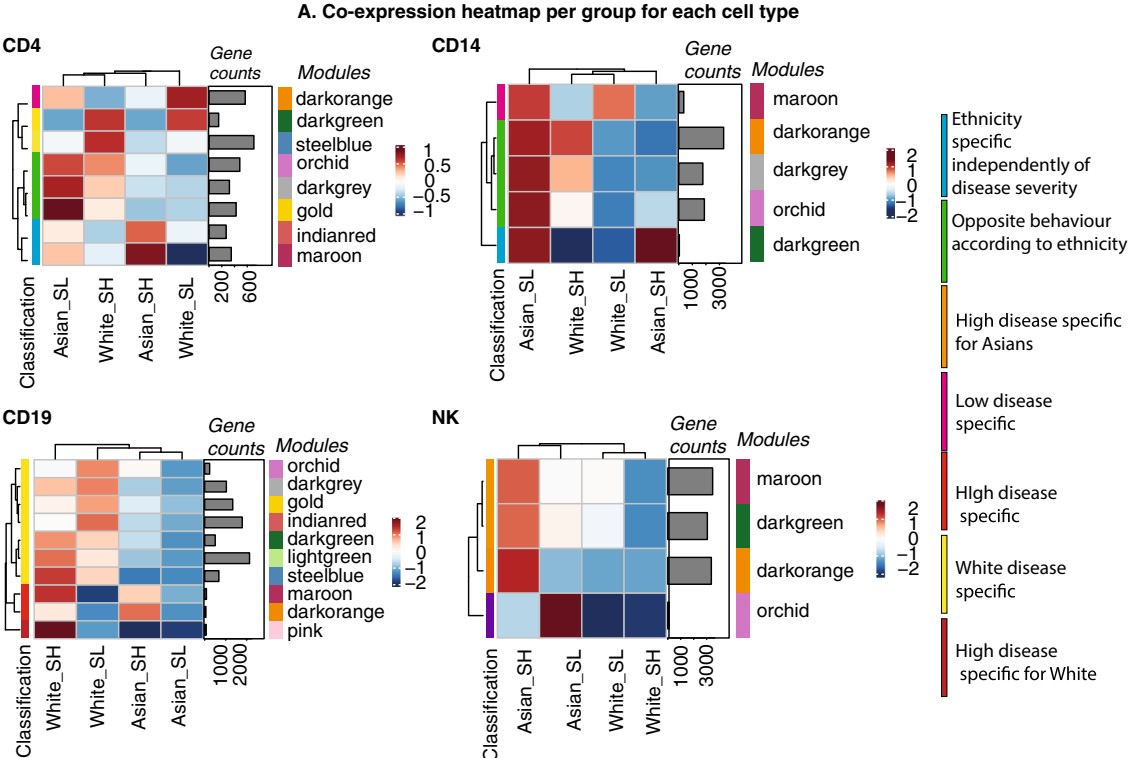

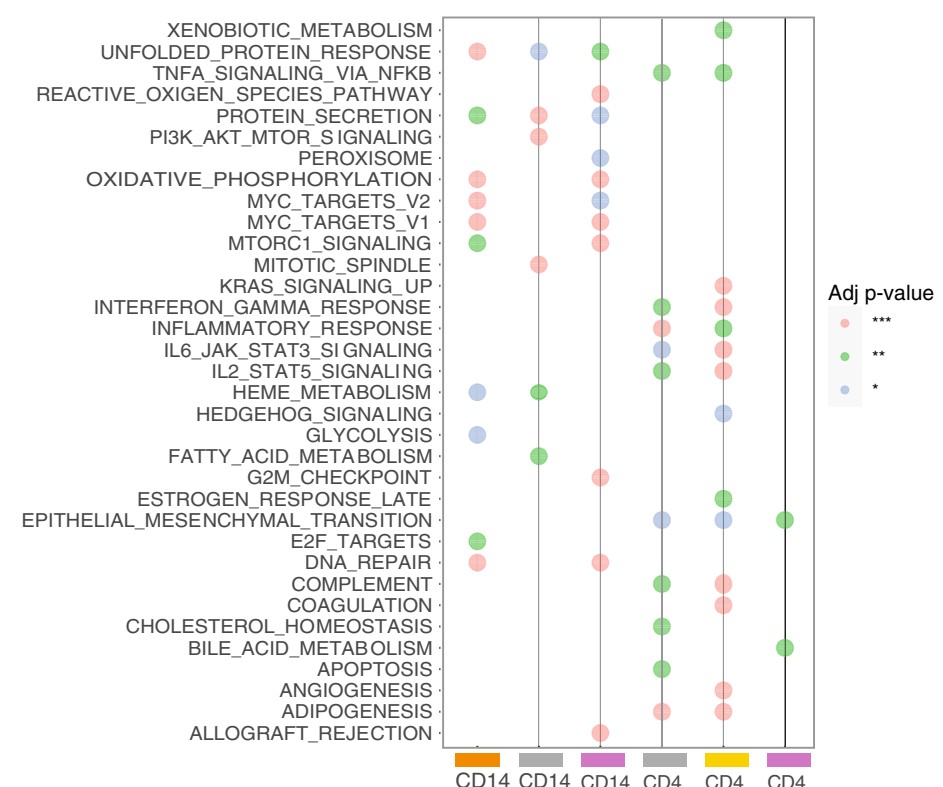

**Fig. 4 Gene co-expression analyses. A** Group fold change heat map and hierarchical clustering for the four data-driven sample groups and the gene modules identified byCoCena2 analysis. Classification: the annotation of the modules into different categories based on their behavior. **B** Functional enrichment of CoCena2-derived modules classified as "Opposite behavior according to ethnicity using the Hallmark gene set database". Only significant terms are visualized.

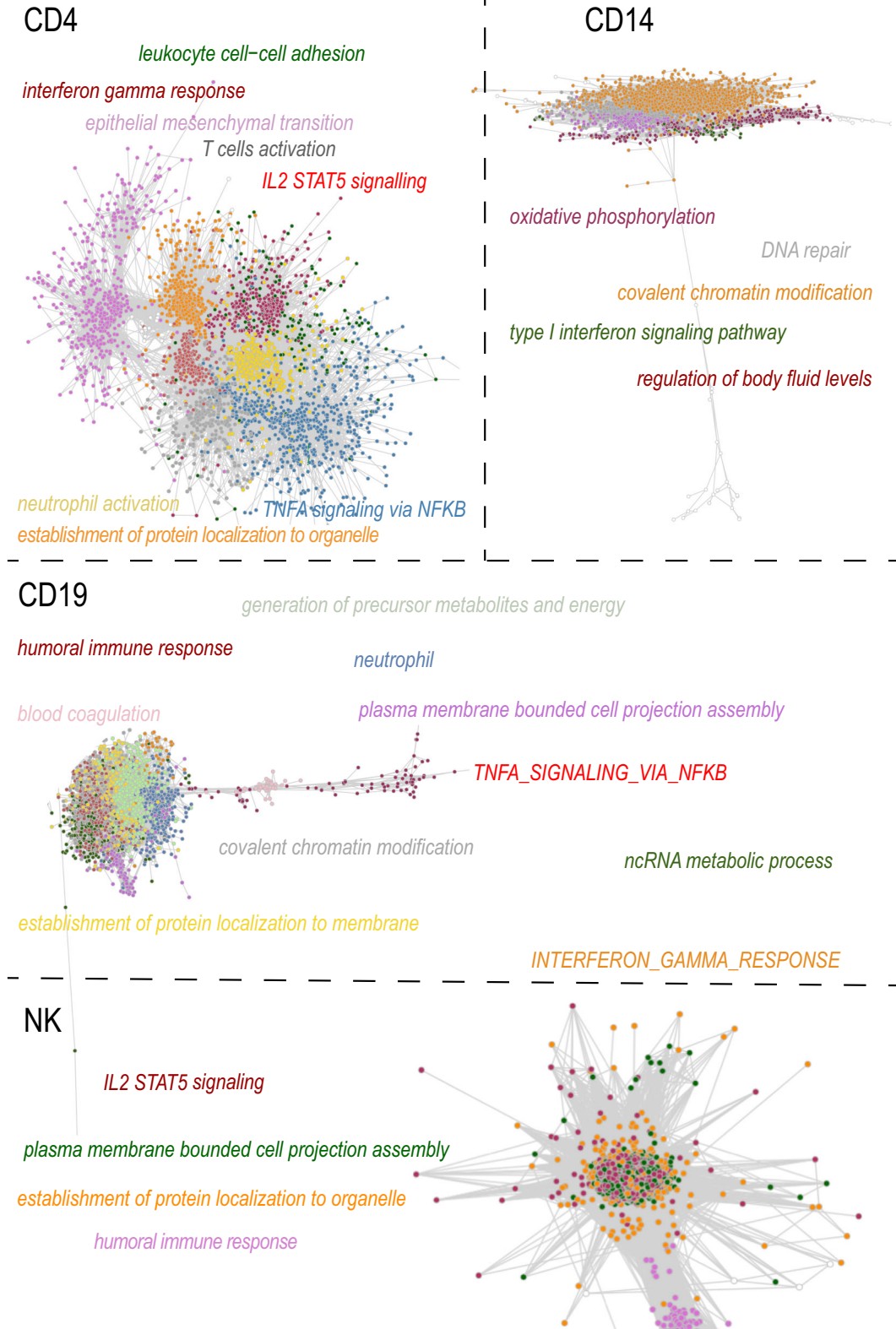

**Fig. 5 Network visualization of gene co-expression analyses.** Visualization of the CD4, CD14, CD19, and NK cells CoCena2 network. Nodes are genes and edges represent co-expressed genes. Additional module information is displayed by module-colored labels. Labels include information about top-GO terms, as well as representative Hallmark terms.

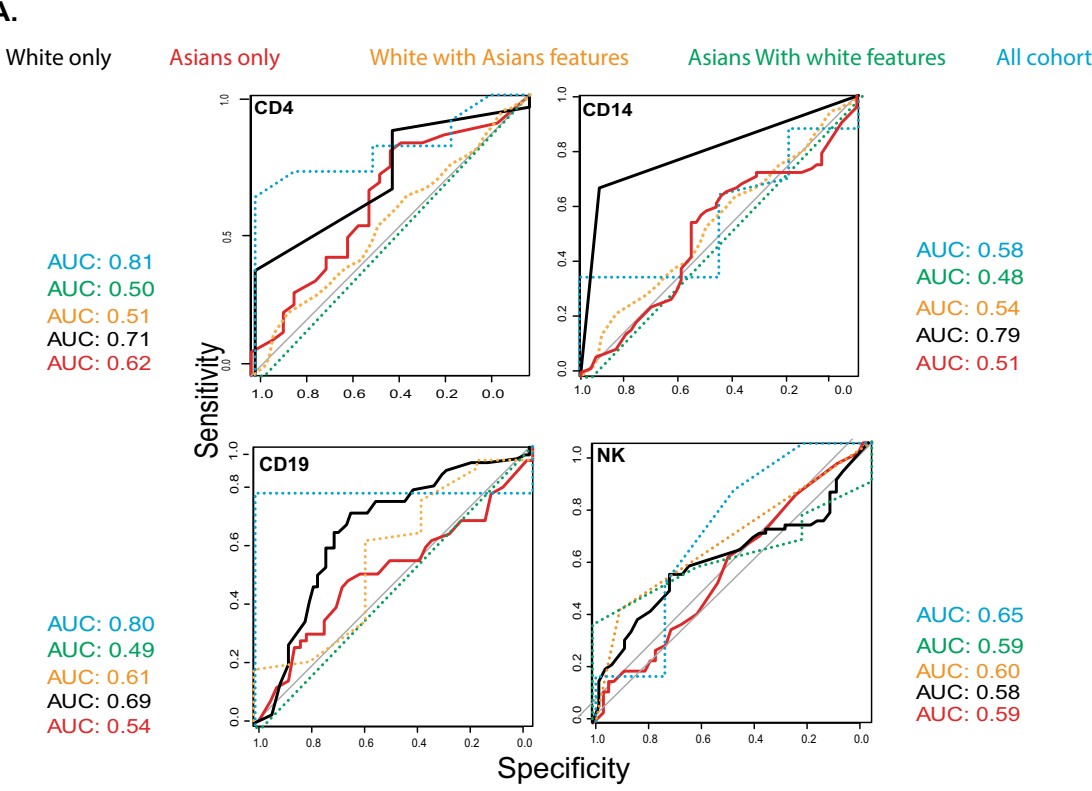

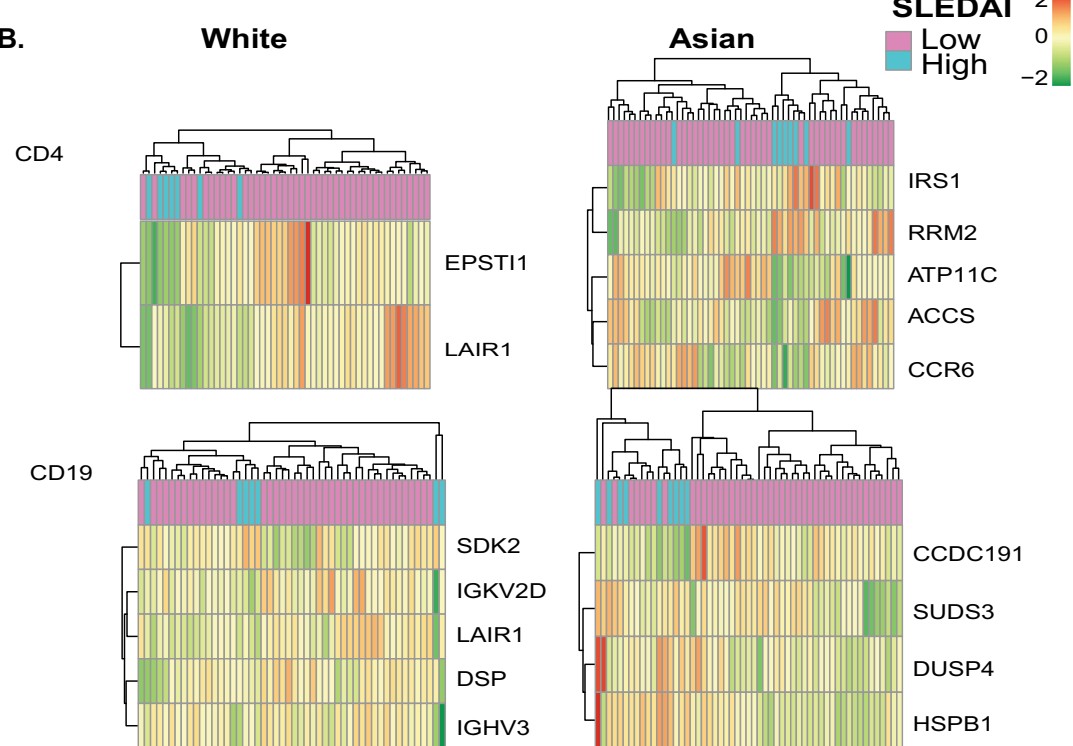

**Fig. 6 Machine learning classifier. A** Area under the ROC curve of machine learning classifier across the White and Asians data sets in discriminating active vs. inactive patients. White (black) and Asians (red) **B** Clustergram generated by using the expression values of the top predictors selected by FSelector function.

CD4$^+$ T cells (Fig. 2C). As shown also on the clinical data work, the Lupus Severity Index, a validated scoring system based on the ACR classification criteria, was also significantly different between the three clusters (Fig. 2D).

Within each immune cell-type, each identified cluster was characterized by specific clinical features (Supplementary Data 4–7). We also observe an opposite association between cluster 1 and cluster 2 of the CD14$^+$ monocytes and ethnicity (Asians and White). To better understand this association, we decided to conduct differential expression analysis to extract the cell-type-specific transcriptional signature expression in SLE patients of ethnically different backgrounds. Supplementary Data 2 lists the significant differentially expressed genes for each cell line. Some of the shared genes across the four cells were involved in SLE or autoimmune-related pathways such as: AFAP1, which was studied in Plasmacytoid dendritic cells in lupus mice[20], USP32P1 which is an Immune-Induced gene-specific of Caucasians only[21], and RPL3P2 which is a gene mapping in the HLA class I region and in the autosomal dominant polycystic kidney disease gene region[22]. The top three upregulated genes in the CD4$^+$ T cells for the White cohort were AFAP1, a gene shown to have an expression pattern similar to the interferon-alpha in SLE mice[20], USP32P1, a ubiquitinase specific of the White population[21], and NAP1L3, Nucleosome Assembly Protein 1 Like 3, which has been recently listed as one of the 93-SLE gene signature (SLE MetaSignature) that is differentially expressed in the blood of patients with SLE compared with healthy volunteers[23]. ARHGEF10, FMN1, and CD79A are the top three significant upregulated genes in CD4$^+$ T cells in Asians. These three genes are specific to the immune system and have been shown to be associated with SLE[24,25]. Two of the top significant upregulated genes in CD14$^+$ monocytes for the White cohort were SNORD3B-2, a gene involved in the biological pathways regulated by progranulin[26], and the lncRNA maternally expression gene 3, MEG3. Long non-coding RNAs (lncRNAs) have emerged as important regulators of biological processes and substantial evidence has been accumulated showing that lncRNAs involved in the pathogenesis of rheumatoid diseases such as SLE[27]. In Asians, Neurotensin Receptor 1 (NTSR1), and RETN, a gene involved in the development of cardiovascular diseases among Egyptian SLE patients[28], were between the significantly upregulated genes in Asians in CD14$^+$ monocytes. Examples of upregulated genes in B cells for the White cohort were ARHGAP24, shown to be upregulated in SLE patients[29], and CHL1 (Cell Adhesion Molecule L1 Like), a protein encoded by this gene is a member of the L1 gene family of neural cell adhesion molecules. Genes encoding for immunoglobulin heavy-chain and light-chain genes were statistically significant upregulated in Asians in B cells. These genes play an important role in antigen recognition[30]. Regarding NK cells, upregulated significant genes in White were SNORD3B-2, and Cytokine Like 1 (CYTL1), while in Asians were HPGD and ADAM19, a gene highly expressed when SLE murine models are subminiaturized with VGX-1027[31]. Gene set enrichment analyses (GSEA) on differentially expressed genes based on SLICC score, SLEDAI score, lupus severity index, and lupus nephritis (Supplementary Fig. 6) showed an elevated expression of interferon pathways in the CD14$^+$ monocytes. We observed an enrichment of interferon pathways in the CD4$^+$ T cells in the differentially expressed analysis conducted on the SLEDAI score.

Analysis of gene expression data has been widely used in transcriptomic studies to understand functions of molecules inside a cell and interactions among molecules, however, differential gene expression analysis might prevent the dissection of the heterogeneity of the disease. For this reason, we conducted co-expression analyses using CoCena taking demographic and clinical features into account. Co-expression analysis identified similarly regulated genes across samples and grouped these genes

into modules. Samples were grouped into White-high, Asians-high (high disease activity defined by a SLEDAI score > =6) and White-low, Asians-low (low disease activity defined by a SLEDAI score < 6). Modules were annotated according to their behaviors into seven categories (Fig. 4A). For example, the Interferon-gamma response pathway was enriched in modules dark gray and gold for CD4$^+$ T cells (Fig. 4B). Extended analysis of the modules revealed a prominent enrichment of inflammatory, autoimmune, and interferon pathways (Fig. 4 and Supplementary Fig. 7). Cocena2 further identified modules that contained transcription factor signature genes known to be associated with SLE. As an example, the top three transcription factors (TF) in the dark orange module of CD4$^+$ T cells are ETS1[32], SP1[33], and ELK4[34], and for the gold module in the CD4$^+$ T cells, the top three TF are Sfpi1-1[35], SPI1[33], and Hand1[36]. We also observe interferon enriched modules in dark gray, gold, Indian red, and maroon (CD4), dark green (CD14), dark orange (CD19 and NK) (Supplementary Fig. 7).

Finally, using an RF approach we were able to extract features within each cell type for each ethnic group that help to discriminate high disease from low disease activity patients. Random forest classification of disease activity in the White and Asian cohorts showed the best classification in CD4$^+$ T cells White cohort with two genes as features (EPSTI1 and LAIR1), which corresponded to an AUC value of 0.71 (Fig. 6). Both genes are involved in the interferon pathway. Epithelial stromal interaction 1 (EPSTI1) is an interferon (IFN) response gene[37], LAIR-1 may play a relevant role in the mechanisms controlling IFNα production by dendritic cells both in normal and pathological innate immune responses[38], IRS1 expression is altered in SLE mice[39], and the frequency of the IGHV3 gene family has been assessed over glucocorticoid time treatment[40]. These were the strongest results observed suggesting that genes expressed by CD4$^+$ T cells may prove to be informative in the study of cell-specific methods of SLE pathogenesis.

Strengths of this study include the rich phenotyping data and one of the largest SLE cohorts including White and Asian patients profiled for cell sorted RNA-seq, which allowed us to identify genetic effects of race in SLE disease activity, shedding light on molecular mediators of race in disease heterogeneity. Future work will include testing these findings in other multi-ethnic cohorts and extending this work beyond transcriptomics to include a multi-omics approach. There are several limitations that should be recognized. Our study is limited by the restriction of transcriptomic data to the one-time point. At the time of blood collection, patients were stable on medications, not currently in lupus flares, and only a few presented with high ($x > 6$) SLEDAI score. In our study, we correlated gene expression with disease activity of the disease at the time of blood draw (SLEDAI), as well as with overall disease manifestations. However, since gene expression is dynamic and may reflect cellular responses to an underlying disease process, it is possible that patients transition between transcriptomic clusters during their disease course. This remains a testable hypothesis in future studies incorporating longitudinal profiling. Although new 2019 classification criteria for systemic lupus erythematosus have recently been published[41], the patients were recruited prior to 2019 and therefore the ACR criteria were used instead. The k-means algorithm used to identify clusters for this analysis could be sensitive to outliers, for this reason, we used PCA to confirm the identified clusters. Specificity and sensitivity were low in all cell types and this could be due to the imbalanced dataset. ML models with substantial predictive accuracy can assist clinicians with complicated diagnostic decision-making, though extensive studies are necessary to construct accurate models for specific population groups. Our study is also limited by the fact that all patients within our cohort are currently

stable on medication making the identification of transcriptomic differences between patients nontrivial, as well as the absence of controls. Although there are large numbers of publicly available gene expression profiles of SLE patients, many of these profiles are not annotated with SLEDAI scores and are not based on cell-sorted data, and thus do not examine specific cell types. Furthermore, some data sets which include SLEDAI scores show heavy class imbalance, which impedes classification. Further work to integrate cross-platform expression data will be crucial to expanding our ability to classify active and inactive SLE patients.

In summary, we have identified distinct clinical subtypes of SLE using immune cell sorted transcriptomic data that have distinct associations with clinical measures, specifically with lupus severity index. We observe a relationship between transcriptomic clusters of CD4$^+$ T cells and the clinical K-means clusters of the same patients. We also identified a positive association between the transcriptomic cluster 1 of the CD14$^+$ monocyte cells and White ethnicity. Pathway enrichment analyses on the differential expressed genes highlighted cell-type-specific upregulated immune pathways for either Asian or White SLE patients including TNF-alpha signaling via NFKB in CD4$^+$ T cells in Whites and IL2 STAT5 signaling in NK cells biopsies in Asians. In addition, we are able to identify modules of genes that are either ethnicity-specific, disease-specific, as well as train machine learning models to discriminate patients based on disease activity within each ethnic group. Our findings provide an insight into the cellular processes that drive SLE pathogenesis within patients of different ethnicity and may eventually lead to customized therapeutic strategies based on patients' unique patterns of cellular activation.

## Methods

**Study design**. Participants were recruited from the California Lupus Epidemiology Study (CLUES). CLUES was approved by the Institutional Review Board of the University of California, San Francisco. All participants signed a written informed consent to participate in the study. Study procedures involved an in-person research clinic visit, which included collection and review of medical records prior to the visit; history and physical examination conducted by a physician specializing in lupus; collection of biospecimens, including peripheral blood for clinical and research purposes; and completion of a structured interview administered by an experienced research assistant. All SLE diagnoses were confirmed by study physicians based upon one of the following definitions: (a) meeting ≥4 of the 11 American College of Rheumatology (ACR) revised criteria for the classification of SLE as defined in 1982 and updated in 1997, (b) meeting 3 of the 11 ACR criteria plus a documented rheumatologist's diagnosis of SLE, or (c) a confirmed diagnosis of lupus nephritis, defined as fulfilling the ACR renal classification criterion (>0.5 grams of proteinuria per day or 3+ protein on urine dipstick analysis) or having evidence of lupus nephritis on kidney biopsy. Based on sample availability at the time of sequencing, a total of 120 patients (Supplementary Data 1) were profiled with bulk RNA-seq from the CLUES cohort. Clinical data collected at sampling and the self-reported race was used for downstream analyses.

**RNA-seq processing and quality control**. Peripheral blood mononuclear cells were isolated from patient donors. Cells were isolated from peripheral blood utilizing magnetic beads (CD14$^+$ monocytes, B cells, CD4$^+$ T cells, and NK cells) using EasySep protocol from STEM cell technologies on 120 patients totaling 480 samples. RNA and DNA were extracted using Qiagen column and quality was assessed on Agilent bioanalyzer. 1–5 ng of RNA was used with SmartSeqv2 protocol to get cDNA, followed by Illumina Nextera XT DNA library prep with the input of 0.8 ng cDNA and 15 cycles of PCR amplification. Both cDNA and DNA library qualities were controlled on an Agilent bioanalyzer. Libraries were sequenced on HiSeq4000 PE150. No statistically significant difference was observed between the distribution of cell counts sorted across groups of interest (Supplementary Fig. 9).

Salmon v0.8.2 was used for our alignment-free pipeline. Adapter-trimmed reads were used as input. We quantified gene expression using raw counts and kept the genes that showed an average FPKM value across the samples >0.5. FastQC (v0.11.8) was run to assess the quality of the sequence reads. To further assess for low-quality samples we checked the expression of 10 housekeeping genes (e.g., *SNRPD3*, *EMC7*, and *VPS29*) called the Eisenberg housekeeping genes[42]. Finally, we conducted K-means clustering as a further quality control step on the whole 480 samples and removed all outlier samples that were > 1 standard deviation from the center of the cluster. The Davies-Bouldin index[43] was also computed to confirm cluster stability. The batch correction was performed with limma (v 3.40.6). Data were adjusted for sequence lane, medications, and sex. A false discovery rate (FDR) of 5% was used as a statistical threshold in all exploratory analyses described below.

**Unsupervised clustering approach and association analyses**. Unsupervised K-means clustering was performed on the batch corrected transcriptomic data per cell type using the factoextra package (v 1.0.7). The number of clusters, $k$, was chosen by maximizing cluster stability measured by Jaccard similarity[44] using a bootstrap resampling-based method. The identified clusters were then tested for associations with disease activity at the time of blood draw (SLEDAI), and clinical and demographic variables such as sex, race, and the ACR criteria. Missing values were excluded from the associations. Fisher's exact test was used to evaluate if categorical variables were enriched in a cluster with respect to the others. For continuous variables, we used ANOVA testing.

**Differential expression analyses**. We performed differential expression gene testing with DESeq2 (v.1.24.0 R package) using default settings. Sequence lane, medications, and sex were used as covariates within the DESeq2 model. Statistical significance was set a 5% FDR (Benjamini-Hochberg). The Bioconductor package fgsea (v 1.10.1), was used for gene set enrichment analysis (GSEA). Differential expression analysis was conducted for several clinical features, specifically: SLEDAI score[45], the mean Systemic Lupus International Collaborating Clinics/American College of Rheumatology (SLICC) Damage Index score[46], lupus severity index, a validated scoring system based on the ACR classification criteria[16], the presence of lupus nephritis, and ethnicity (White vs. Asians), Supplementary Figs. 2–6.

**Sensitivity analyses**. Some patients recruited in the cohort did not satisfy at least 4 1997 ACR criteria ($n = 9$) or did not have a biopsy confirmed diagnosis of lupus nephritis ($n = 12$). Although in all patients lupus and lupus nephritis diagnoses were confirmed by a national lupus expert based on a thorough in-person review clinical visit that included patient's history, physical examination, laboratories, and medical records, we conducted sensitivity analyses excluding these patients ($n = 21$) and confirmed that our clustering and differential expression findings were robust (Supplementary Figs. 11 and 12).

**Co-expression analysis**. The CoCena2 pipeline [https://github.com/Ulas-lab/CoCena2] was used to define differences and similarities in transcript expression patterns between the two ethnic groups and disease activity (based on the SLEDAI score). Patients were divided into White-high, Asians-high (high disease activity defined by a SLEDAI score > =6) from White-low, Asians-low (low disease activity defined by a SLEDAI score <6) for each cell type. As previously described in other publications, CoCena2 (Construction of co-expression Network Analysis–automated) was performed based on Pearson correlation. All genes were used as the input. Data were adjusted for sequence lane, medications, and sex. Pearson correlation was performed using the R package Hmisc (v4.1-1). To increase data quality, only significant (p < 0.05) correlation values were kept. The nodes were colored based on the Group Fold Change (GFC), the mean of each condition versus the overall mean for each gene, respectively, for each group separately. Unbiased clustering was performed using the "label propagation" algorithm in the graph (v1.2.1) and was repeated 1000 times. Genes assigned to more than 5 different clusters during the iterations received no cluster assignment. The mean GFC expression for each cluster and condition was visualized in the Cluster/Condition heatmap. Clusters smaller than 10 genes were not shown[47].

**Machine learning**. Prior to applying machine learning, gene selection and normalization were performed using the R packages DaMIRSEq. (v1.2.0) and MLSeq (v 2.2.1). Specifically, data were adjusted for sequence lane, medications, and sex. Accuracy of seven different ML algorithms was evaluated prior to choosing the Random Forrest model (Supplementary Fig. 10). Within each cell type, the data was split into training and testing set at 70% for training and 30% for testing in order to distinguish high disease activity (SLEDAI score > = 6) samples from low disease activity samples (SLEDAI score <6) in both the Asian and White populations. To prevent the inclusion of redundant features that may decrease the model performance during the classification step, a function that produces a pair-wise absolute correlation matrix was applied. The FSelector package was used to rank features. Default parameters were used for all packages. For each random forest model, we calculated the mean area under the curve (AUC) over ten cross-validation folds. As cross-validation, features identified in the White cohort for a specific cell type were used to train the model on the Asian cohort for the same cell type and vice versa. Mean AUC was assessed over ten cross-validation folds.

**Statistics and reproducibility**. A total of 120 patients (Supplementary Data 1) were profiled with bulk RNA-seq from the CLUES cohort with no replicates. The statistical analyses for each part of the approach are described above and the code is available to ensure the reproducibility of our results (https://rb.gy/mhyafy).

**Reporting summary**. Further information on research design is available in the Nature Research Reporting Summary linked to this article.

## Data availability
Source data underlying main figures are presented in Supplementary Data 1–8, and the raw data of this study are openly available in GEO: GSE164457. All other data are available from the corresponding author upon reasonable request.

## Code availability
R custom code used to generate the figures and analysis can be found on figshare https://rb.gy/mhyafy.

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

## Acknowledgements
We are grateful to all of the patients who participated in the study. We are also grateful to Ms. Bushra Samad, and Dr. Ellie Seaby who provided advice on QC methods and discussion of study results. This study was funded through the following grants: P30 AR070155 (M.S., G.A., L.A.C.), P60AR053308 (L.A.C.), K01LM012381 (M.S.), F32 AR070585 NIAMS (M.G.), U01DP005120 CDC (L.A.C., C.M.L., J.Y., M.D., L.T., P.K.), the Rheumatology Research Foundation 128849A (C.M.L.), and the Lupus Research Alliance (L.A.C.).

## Author contributions
G.A.: study design, data analysis, quality control analyses, data interpretation, and paper preparation. C.M.L., L.T., I.P., T.S.J., A.J.C., C.J.Y.: data interpretation, and paper

revision. C.J.Y., J.N., K.E.T., L.M.: data generation, quality control analyses, genetic ancestry estimates, and paper revision. P.K., M.D., J.Y.: CLUES co-investigators. Patient enrollment and clinical, demographic characterization of patients, and paper revisions. L.A.C., L.T.: data generation, study design, data interpretation, and paper revisions. M.S.: Computational expertise, study design, data interpretation, and paper revisions. All authors edited and critically revised the manuscript for important intellectual content and gave final approval for the version to be published.

## Competing interests

The authors declare no competing interests.
