## [Peer Review File · Communications Biology]

Reviewers' comments:

Reviewer #1 (Remarks to the Author):

The Authors utilise transcriptomics data (RNA-seq) from purified immune cell subsets from a bi-ethnic SLE cohort (Asian, White) in order to discern ethnic-relevant/specific clusters. This analysis is relevant considering the ethnic variation in disease phenotype, prognosis and response to treatments.

Major point:

- As the authors discuss in the Introduction, further suggested by their results, gene expression variation may be confounded by difference in the clinical manifestations of lupus in White vs. Asian groups. Have the data been appropriately adjusted?
- How was ethnicity determined? Was it self-reported? Were the subjects genotyped for tag-SNPs corresponding to the ethnic background?

Minor points:

- Abstract. "Samples were grouped into groups base on their disease...". Please correct.
- Abstract. Generally, the flow of the results needs improvement as it is hard for the reader to follow the sequence and rationale of the comparative analyses.
- ACR criteria: please correct "18" with "11".
- 4 cell subtypes x 2 Ethnic groups = 8 groups. Was FDR 5% used as a statistical threshold in all exploratory analyses?

Reviewer #2 (Remarks to the Author):

This is a comprehensive analysis of the transcriptome of specific immune cell types from the peripheral blood of patients with SLE.

This a superb effort to characterize the immune response in SLE. The work is thorough and very detailed.

My main concern is in regards to the stability of the transcriptomic phenotype overtime.

Most of the correlative studies done here are between the specific transcriptomic phenotype and the overall profile of the patient, ie the clinical and laboratory manifestations over their disease course and not manifestations at the time of the blood draw. I am not sure we can assume that the transcriptomic profile of the patients stays the same over time.

The other concern I have is in regards to the total number of cells that is not taken into account. Patients with SLE are known to have variability in the number of monocytes and lymphocytes while neutropenia and/or lymphopenia can be related to medications. Actually Medication effect is not taken into account at all.

Specific comments:

Methods: The participants included in this study may not be fulfilling the ACR criteria (some may only fulfill 3 out of 11 and a rheumatologist's diagnosis). That is not standard in studies of SLE. The authors should use the new criteria (EULAR/ACR 2019) or use the ACR 1997 criteria.

Fig 2. The most important finding is correlation with disease activity (SLEDAI). As above the importance of the correlation with ACR criteria manifestations, which may not be currently active, may not be as high. Correlation with the new weighted EULAR/ACR 2019 criteria instead of the lupus

severity index may be more relevant.

In D the authors should use pairwise non parametric analysis.

Fig 3. Are the differences between races explained by the differences in disease activity or disease manifestations?

Fig 6. The SLEDAI in the cohort are generally very low making the interpretation of the data presented here very difficult. Medication effect on gene expression is not taken into account.

Reviewer #3 (Remarks to the Author):

Comments

The authors use CD14+ monocytes, B cells, CD4+T cells and NK cells RNA-seq data from SLE patients in order to stratify patients based on their gene expression signatures. They employ machine learning, unsupervised clustering, differential expression analyses and gene co-expression analyses. They report distinct clinical subtypes of SLE using transcriptomic data from immune cells. They report gene clusters that are ethnicity specific, disease specific and use ML models to separate patients based on disease activity within each ethnic group. This work appears to be of interest and has potential contribution to the current literature shedding light towards the direction of customised therapeutic strategies based on the unique patterns of cellular processes in each patient. In total, the research question and the findings are interesting, but there are limitations the authors need to address and provide clarifications.

- How did the authors handle the missing values?
- Reference for Jaccard score
- It would be interesting to see another test, such as Silhouette values and the relative plots, besides the Jaccard score and how they compare.
- In the methods section, lines 412-413. "Finally, we conducted K-means clustering and removed all outlier samples that were > 1 standard deviation from the centre of the cluster." Then, at the legend of Figure 1 they refer: "After removal of the outliers/ low quality samples we used three approaches: unsupervised approach, a more traditional approach and a supervised approach.". The authors need to clarify this section. What was their procedure?
- Why did the authors choose K-means? Did the authors test k-mode as it could be more appropriate for this data?
- Why the authors did not consider to use and/or another classification approach, such as agglomerative hierarchical clustering with some other distance (not Eukleidian)?
- What are the parameters the authors used for the RF?
- Why did the authors choose RF and/or not another (ML) approach? Such as logistic regression and/or Support Vector Machine?
- Figure 1. How and why were the 120 out of the 400 patients were selected?
- Table S5, S6, S7 the legend has twice the same sentence.
- Figure 6. They refer at the legend White (black) and Asians (red) and at the graph the colours are opposite

We would like to thank the editors and reviewers for the thoughtful in depth review of our work. Please find the point by point responses to the reviewer comments below in bold. We would like to point out to the editorial team that the data availability statement of the study has been added on page 20.

Reviewers' comments:

Reviewer #1 (Remarks to the Author):

The Authors utilise transcriptomics data (RNA-seq) from purified immune cell subsets from a bi-ethnic SLE cohort (Asian, White) in order to discern ethnic-relevant/specific clusters. This analysis is relevant considering the ethnic variation in disease phenotype, prognosis and response to treatments.

We would like to thank the reviewer for summarizing our work and the constructive feedback. Please see our point by point responses below.

Major point:

- As the authors discuss in the Introduction, further suggested by their results, gene expression variation may be confounded by difference in the clinical manifestations of lupus in White vs. Asian groups. Have the data been appropriately adjusted?

We thank reviewer for raising this point. In Table S1 we describe the demographics of the cohort split by ethnic group. As it is visible by the data, the two groups do not present a significant difference in terms of disease activity, indicated with the SLEDAI score (mean score in Asian and White of 2.7 and 2.5 respectively, p-value of 0.808). Therefore, we don't believe an adjustment based on the clinical manifestation is necessary given the clinical similarities between the two groups. We believe this similarity could be explained by the fact that at the time of sample collection and sequencing patients were currently stable and we have indicated this as a limitation of the study on page 16.

- How was ethnicity determined? Was it self-reported? Were the subjects genotyped for tag-SNPs corresponding to the ethnic background?

We confirm with the reviewer that ethnicity was self-reported. Patients were not genotyped for tag-SNPs corresponding to the ethnic background. We have also clarified this on page 19 of the Methods section of the manuscript.

Minor points:

- Abstract. "Samples were grouped into groups base on their disease...". Please correct.

Thank you this has been corrected in "*Samples were grouped into groups based on their disease...*"

- Abstract. Generally, the flow of the results needs improvement as it is hard for the reader to follow the sequence and rationale of the comparative analyses.

We thank reviewer for the comments, we reworded the results section of the abstract to improve flow.

- ACR criteria: please correct "18" with "11".

Thank you, this has been corrected in “*In our own previous work, unsupervised clustering of the 11 American College of Rheumatology (ACR)*”

- 4 cell subtypes x 2 Ethnic groups = 8 groups. Was FDR 5% used as a statistical threshold in all exploratory analyses?

Yes, we confirm a FDR of 5% has been used in all analyses. We have stated this more clearly at the beginning of the Methods section instead of within each analyses paragraph as we have done previously on page 20: “*Batch correction was performed with limma (v 3.40.6). A false discovery rate (FDR) of 5% was used as a statistical threshold in all exploratory analyses described below.*”

Reviewer #2 (Remarks to the Author):

This is a comprehensive analysis of the transcriptome of specific immune cell types from the peripheral blood of patients with SLE. This a superb effort to characterize the immune response in SLE. The work is thorough and very detailed.

We would like to thank the reviewer for recognizing the value of our work.

My main concern is in regards to the stability of the transcriptomic phenotype overtime. Most of the correlative studies done here are between the specific transcriptomic phenotype and the overall profile of the patient, ie the clinical and laboratory manifestations over their disease course and not manifestations at the time of the blood draw. I am not sure we can assume that the transcriptomic profile of the patients stays the same over time.

Thank you for pointing out this particular issue. We confirm that the clinical data used in these analyses have been collected at the time of blood draw and this has now been specified in the Methods section (page 19). Unfortunately, the study design here only allows for a single sample (snapshot) molecular measurement. We do not assume that the transcriptomic profile of the patients stays the same over time and we stated this as a limitation of this study on page 16: “*There are several limitations that should be recognized. Our study is limited by the restriction of transcriptomic data to one time point. At the time of blood collection, patients were not currently in lupus flares. However, since gene expression is dynamic and may reflect cellular responses to an underlying disease process, it is possible that patients transition between transcriptomic clusters as during their disease course.*”

The other concern I have is in regards to the total number of cells that is not taken into account. Patients with SLE are known to have variability in the number of monocytes and lymphocytes while neutropenia and/or lymphopenia can be related to medications.

Thank you so much for bringing up this important issue. We would like to point out that the data we use is cell sorted data, not single cell data, and approximately the same number of cells (10^6) were extracted in the same patients preventing the loss of a specific cell subset. We have added the distribution of cell count sorted for each cell type split across the transcriptomic clusters in Asian and White cohorts as supplementary figure S9. Non-parametric test shows no significant difference between the distributions across the groups. We have also added a sentence about this on page 20 of the Methods section.

Actually Medication effect is not taken into account at all.

Also we confirm that sequence lane, medications, and sex were used as covariates within the DESeq2, co-expression model and ML models. This has now been made clearer in the Methods section on page 20.

Specific comments:

Methods: The participants included in this study may not be fulfilling the ACR criteria (some may only fulfill 3 out of 11 and a rheumatologist's diagnosis). That is not standard in studies of SLE. The authors should use the new criteria (EULAR/ACR 2019) or use the ACR 1997 criteria.

We would like to thank the reviewer for this suggestion. It's important to note that the classification criteria were not meant for diagnosis and are not used for clinical care. In all of these patients, lupus diagnoses were confirmed by a national lupus expert based on a thorough in-person review of the patient's history, physical examination, laboratories and medical records (Drs. Dall'Era, Yazdany or Lanata). In our study, most patients met classification criteria, and those that did not were still felt to have definite SLE. This is expected as classification criteria do not have perfect sensitivity nor specificity. We appreciate the suggestion to use the new criteria from 2019, but unfortunately not all the clinical variables defined in the EULAR/ACR 2019 criteria were collected as part of this effort because the patients were recruited prior to 2019. Therefore, we choose to use ACR criteria, but we have added a sentence to the discussion paragraph describing this issue on page 17.

Fig 2. The most important finding is correlation with disease activity (SLEDAI). As above the importance of the correlation with ACR criteria manifestations, which may not be currently active, may not be as high. Correlation with the new weighted EULAR/ACR 2019 criteria instead of the lupus severity index may be more relevant.

We would like to thank the reviewer for raising this question. Since the EULAR/ACR 2019 criteria are not an index with a continuous score (instead there are cut points for SLE vs. not) and have not been shown and validated to correlate with SLE activity/severity, we have chosen to use the lupus severity index (Bello, G. A. *et al.* Development and validation of a simple lupus severity index using ACR criteria for classification of SLE. *Lupus Sci. Med.* 2016), which is a continuous variable and has been validated. Also, unfortunately, not all the clinical variables defined in the EULAR/ACR 2019 criteria were collected as part of this effort because the patients were recruited prior to 2019. Therefore, we choose to use ACR criteria, but we have added a sentence to the limitations paragraph describing this drawback on page 17.

In D the authors should use pairwise non parametric analysis.

We would like to thank the reviewer for bringing up this point, we repeated the analyses using Kruskal-Wallis and the Wilcoxon tests. Results have more conservative p-values but still in line with our previous findings. The new p-values have now been added to Figure 2D on page 32.

Fig 3. Are the differences between races explained by the differences in disease activity or disease manifestations?

This is a very interesting comment and is in line with the feedback from Reviewer 1. In Table S1 we describe the demographics of the cohort split by ethnic group. As it is visible by the data, the two groups do not present significant differences in term of disease activity as described by the SLEDAI score (mean score in Asian and White of 2.7 and 2.5 respectively, p-value of 0.08). Even in terms of flares manifestation of lupus nephritis or presence of anti-double stranded DNA antibodies we do not observe significant differences between groups. This might reflect one limitation of this study, discussed on page 16, regarding patients being stable on medications.

Fig 6. The SLEDAI in the cohort are generally very low making the interpretation of the data presented here very difficult.

Thank you for bringing up this limitation of our study. Unfortunately, patients selected in this research cohort have a low SLEDAI score (mean < 3) and we added this point as a limitation of the study on page 16. However, we still wanted to decipher differences at the transcriptional level even in stable lupus patients. Although we have a small sample size, our machine learning model was successfully able to distinguish patients based on their disease activity within each ethnic group.

Medication effect on gene expression is not taken into account.

We apologize that this wasn't clear before, but sequence lane, medications, and sex were used as covariates within the model, which we have now clarified in the Methods section on page 20.

Reviewer #3 (Remarks to the Author):

Comments

The authors use CD14+ monocytes, B cells, CD4+T cells and NK cells RNA-seq data from SLE patients in order to stratify patients based on their gene expression signatures. They employ machine learning, unsupervised clustering, differential expression analyses and gene co-expression analyses. They report distinct clinical subtypes of SLE using transcriptomic data from immune cells. They report gene clusters that are ethnicity specific, disease specific and use ML models to separate patients based on disease activity within each ethnic group. This work appears to be of interest and has potential contribution to the current literature shedding light towards the direction of customised therapeutic strategies based on the unique patterns of cellular processes in each patient. In total, the research question and the findings are interesting, but there are limitations the authors need to address and provide clarifications.

We would like to thank the reviewer for a great summary of our work. Please find the point by point responses below.

- How did the authors handle the missing values?

We thank reviewer for this point, we were lucky enough that the clinical data used in the diff expression and co-expression models did not present any missing value. On the other hand, for the correlation analyses, missing values were excluded. This is now clarified on page 20 of the Methods.

- Reference for Jaccard score

Thank you for pointing this out, the Jaccard score reference (Hennig, C. (2007). Cluster-wise assessment of cluster stability. *Computational Statistics & Data Analysis*) has been added on page 20.

- It would be interesting to see another test, such as Silhouette values and the relative plots, besides the Jaccard score and how they compare.

We are happy to share these plots with the reviewer, please see below. As the reviewer might notice all of them are consistent between the two approaches except CD14 monocytes (Fig A-D). The silhouette for the CD14 monocytes (B) shows four as optimal number of clusters, while the Jaccard score shows an optimal number of clusters as 3. After having plotted the dimensionality reduction of the data by the proposed clusters (E), we have decided to follow the Jaccard score recommendation as clusters 3 and 4 overlapped.

A

B

C

D

E

• In the methods section, lines 412-413. “Finally, we conducted K-means clustering and removed all outlier samples that were > 1 standard deviation from the centre of the cluster.” Then, at the legend of Figure 1 they refer: “After removal of the outliers/ low quality samples we used three approaches: unsupervised approach, a more traditional approach and a supervised approach.”. The authors need to clarify this section. What was their procedure?

We apologize for the confusion. This has now been clarified in the text on page 20. The k-means clustering step on the whole cohort helped to remove outliers (as shown by the PCA in supplementary figure 1B). We modified the sentence as the following:

“Finally, we conducted K-means clustering as a further quality control step on the whole 480 samples and removed all outlier samples that were > 1 standard deviation from the center of the cluster.”

- Why did the authors choose K-means? Did the authors test k-mode as it could be more appropriate for this data?

We thank the reviewer for raising the point. We have decided to use k-means clustering as it is one of the most used clustering algorithms for continuous variables. K-means allows us to group the patients according to their existing similarities into k clusters. This enabled us to observe the global profiles of the patients (as previously published in *Nature Comm* on the same cohort in 2019 - <https://doi.org/10.1038/s41467-019-11845-y>). K-mode, on the other hand, is another unsupervised clustering approach based on categorical data (Huang, Z. Extensions to the k-Means Algorithm for Clustering Large Data Sets with Categorical Values. *Data Mining and Knowledge Discovery*, 1998). Differently from k-means, k-mode uses dissimilarities and instead of means it uses modes. Given that we use genes counts derived from RNA sequencing of cell sorted data and that our cohort is composed of cases only whose disease is currently not active (SLEDAI < 3), we deemed k-means was more appropriate to answer our hypothesis rather than another classification approach.

- Why the authors did not consider to use and/or another classification approach, such as agglomerative hierarchical clustering with some other distance (not Eukleidian)?

We would like to thank the reviewer for raising this question. Both hierarchical clustering and k-means are unsupervised learning tasks which have the similar goal. We have decided to use K-means as we wanted to observe the global profiles of the patients (to be consistent with our previously published work in *Nature Comm* on the same cohort in 2019 - <https://doi.org/10.1038/s41467-019-11845-y>).

- What are the parameters the authors used for the RF?

We apologize for leaving out these details initially. For the RF model we applied the DaMiRseq package (Data Mining for RNA- Seq data: normalization, feature selection and classification) and the MLSeq R package (Machine Learning Interface to RNA-Seq Data). Feature selection was done using the FSelector package. Default parameters for all packages were used. We modified the methods to better reflect this on page 22.

- Why did the authors choose RF and/or not another (ML) approach? Such as logistic regression and/or Support Vector Machine?

We would like to thank the reviewer for raising this point, we did try several ML approached before choosing RF. As it is shown from the plots on the newly added supplementary figure 10, we compared accuracies for seven different classifiers and we deemed RF to be the most optimal one for the data. This has now be added in the Methods section on page 22.

- Figure 1. How and why were the 120 out of the 400 patients were selected?

We would like to thank the reviewer for bridging up this question. The 120 samples were selected based on sample availability. We have included a sentence on this in the methods section on page 19.

“Based on sample availability at the time of sequencing, a total of 120 patients (Table S1) were profiled with bulk RNA-seq from the CLUES cohort.”

- Table S5, S6, S7 the legend has twice the same sentence.

We apologize for this oversight. *“Flare severity measurement: mild 1, moderate 2, severe 3”* has been corrected.

- Figure 6. They refer at the legend White (black) and Asians (red) and at the graph the colours are opposite

Thank you so much for catching this error. We apologize and have now corrected this on page 40.

Reviewers' comments:

Reviewer #1 (Remarks to the Author):

The Authors have addressed all points raised by the referees and I am content with the revised version.

Reviewer #2 (Remarks to the Author):

The authors responded to some of my comments.

Nevertheless

-I do not think that patients who fulfill <4 ACR 1997 (the bare minimum for diagnosis) should be included. Moreover patients with proteinuria without a renal biopsy are not definite lupus nephritis patients.

-If the transcriptome is a snapshot it should be correlated with the activity of the disease at the time of blood draw (SLEDAI). If there is correlation of the transcriptome with manifestations of lupus that the patient had at some other point in their disease course, that is interesting but not as important.

Reviewer #3 (Remarks to the Author):

The authors had a very fast turnaround and addressed most of the initial comments. I have some additional comments/questions for the authors to consider.

1. Why did the authors use clustering to identify the outliers at the 400+ samples since they had sample availability for 120?
2. k-means is sensitive to outliers, they should consider mentioning that in the disadvantages and limitations.
3. The Davies-Bouldin's index needs reference - page 20
4. Figure 1 the legend "After removal of the outliers/ low quality samples we used three approached" needs to change to approaches

We would like to thank the editors and reviewers for the thoughtful in depth second review of our work. Please find the point by point responses to the reviewer comments below in bold.

Reviewers' comments:

Reviewer #1 (Remarks to the Author):

The Authors have addressed all points raised by the referees and I am content with the revised version.

We would like to thank the reviewer for contributing to our work and the constructive feedback received.

Reviewer #2 (Remarks to the Author):

The authors responded to some of my comments.

Nevertheless

I do not think that patients who fulfill <4 ACR 1997 (the bare minimum for diagnosis) should be included. Moreover patients with proteinuria without a renal biopsy are not definite lupus nephritis patients.

We would like to thank the reviewer for raising this point. Over the past several weeks, our team carried out extensive chart reviews of patient clinical records to confirm ACR criteria. Nine patients of the 120 examined did not satisfy at least 4 1997 ACR criteria. However, in all of these patients, lupus diagnoses were confirmed by a national lupus expert based on a thorough in-person clinical visit that included patient's history, physical examination, laboratories and medical records (Drs. Dall'Era, Yazdany or Lanata). For the subset of lupus nephritis patients (n=54), 35 had a confirmed biopsy diagnosis, 7 had a clinical diagnosis because the biopsy was contraindicated and 12 patients had no determinative evidence in their records of lupus nephritis but the study physician determined a diagnoses of lupus nephritis was highly likely based on a review of the patient's history and treatment (e.g. a young patient with lupus nephritis diagnosed in China who had received cyclophosphamide therapy but for whom a biopsy report was not available). Based on these observations, we carried out a sensitivity analysis removing these 21 patients (9 with fewer than 4 ACR criteria and 12 with unconfirmed LN), and we were able to show that our results were consistent with our original findings. K-means clustering revealed the same number of clusters, association analyses highlighted the same associations observed previously as well as a new association for cluster 3 of the CD14 monocytes with ethnicity. This new finding makes our previously observed association between the transcriptomic program and monocytes stronger. Differential expression analyses identified the same pathways. We have added this analysis to the Methods section of the paper on page 21 and the new figures as supplementary Figure S11 and Figure S12 (also shown below).

A. K-means clustering

B. Correlation analyses

C. Alluvium plot - clinical and transcriptomic clusters

D.

Figure S11: Sensitivity analyses without patients with ACR criteria < 4 and a confirmed diagnoses of lupus nephritis (n=21): A) K-means clustering on the CD14⁺ monocytes, CD4⁺ T-cells, B cells and NK cells. B) Association analyses between clinical parameters and K-means s clustering. Red – positive

association, blue – negative association. Number of stars indicate the level of significance. C) Alluvium plot visualizing the distribution of the samples according to different clusters. D) Distribution of lupus severity index and SLEDAI score across clusters with p-value computed using non-parametric tests. Findings between the sensitivity analysis and the main analysis are consistent.

Figure S12: Sensitivity analyses without patients with ACR criteria < 4 and a confirmed diagnoses of lupus nephritis: A) Identification of the common DE genes ($p_{adj} < 0.05$ & $abs \log_2FC > 1$) for White

vs Asian cohorts across the different cell types. B) Pathway enrichment analyses for the associated genes in White vs Asian cohorts. Findings between the sensitivity analysis and the main analysis are consistent.

-If the transcriptome is a snapshot it should be correlated with the activity of the disease at the time of blood draw (SLEDAI). If there is correlation of the transcriptome with manifestations of lupus that the patient had at some other point in their disease course, that is interesting but not as important.

We would like to thank the reviewer for bringing up this important point. We would like to note that the SLEDAI score and activity was included in our correlation analysis across the transcriptomic clusters and we observed that it was significantly associated with the clusters for CD4 cells as shown below and presented in Fig 2B, page 33. We now added a specific subpanel highlighting the SLEDAI associations as part of Fig 2D. This association also held up in our sensitivity analysis. (Figure S11). We also included several sentences in the discussion talking about the fact that transcriptomics allows us to capture a snapshot which we correlated with disease activity of the disease at the time of blood draw (SLEDAI) as well as with overall disease manifestations (ACR, etc) on page 16 and on the Methods on page 20.

Reviewer #3 (Remarks to the Author):

The authors had a very fast turnaround and addressed most of the initial comments. I have some additional comments/questions for the authors to consider.

1. Why did the authors use clustering to identify the outliers at the 400+ samples since they had sample availability for 120?

We would like to thank the reviewer for raising this question and apologize for any confusion. Because we are looking at 4 cell types across 120 individuals, that makes 480 samples. We leveraged the k-Means clustering algorithm to make sure the QC is consistent with 4 different cell types. This was done for QC only before running the analysis on each cell type separately to correlate with various clinical and demographic factors. We have made this clear on page 19 of the Methods section.

2. k-means is sensitive to outliers, they should consider mentioning that in the disadvantages and limitations.

We would like to thank the reviewer for raising this point. We have now added the disadvantages and limitations of k-means clustering in the Discussion section on page 17.

3. The Davies-Bouldin's index needs reference - page 20

Thank you for pointing this out, the Davies-Bouldin's index (*Davies D. and Bouldin D., "A Cluster Separation Measure", IEEE Trans. Pattern Anal. Mach. Intell., Vol. 1, No 2, 1979, pp. 224-227*) has been added on page 20.

4. Figure 1 the legend "After removal of the outliers /low quality samples we used three approached" needs to change to approaches.

We apologize for this oversight. "After removal of the outliers/ low quality samples we used *three approaches*" has been corrected.